# What Makes GPCRs from Different Families Bind to the Same Ligand?

**DOI:** 10.3390/biom12070863

**Published:** 2022-06-21

**Authors:** Kwabena Owusu Dankwah, Jonathon E. Mohl, Khodeza Begum, Ming-Ying Leung

**Affiliations:** 1Computational Science Program, The University of Texas at El Paso, El Paso, TX 79968, USA; jemohl@utep.edu; 2Bioinformatics Program, The University of Texas at El Paso, El Paso, TX 79968, USA; kbegum@utep.edu; 3Department of Mathematical Sciences, The University of Texas at El Paso, El Paso, TX 79968, USA; 4Border Biomedical Research Center, The University of Texas at El Paso, El Paso, TX 79968, USA

**Keywords:** GPCR, ligand, binding pocket, 3D structure, motif, docking, binding pose, conformation, GPCR ligand interaction

## Abstract

G protein-coupled receptors (GPCRs) are the largest class of cell-surface receptor proteins with important functions in signal transduction and often serve as therapeutic drug targets. With the rapidly growing public data on three dimensional (3D) structures of GPCRs and GPCR-ligand interactions, computational prediction of GPCR ligand binding becomes a convincing option to high throughput screening and other experimental approaches during the beginning phases of ligand discovery. In this work, we set out to computationally uncover and understand the binding of a single ligand to GPCRs from several different families. Three-dimensional structural comparisons of the GPCRs that bind to the same ligand revealed local 3D structural similarities and often these regions overlap with locations of binding pockets. These pockets were found to be similar (based on backbone geometry and side-chain orientation using APoc), and they correlate positively with electrostatic properties of the pockets. Moreover, the more similar the pockets, the more likely a ligand binding to the pockets will interact with similar residues, have similar conformations, and produce similar binding affinities across the pockets. These findings can be exploited to improve protein function inference, drug repurposing and drug toxicity prediction, and accelerate the development of new drugs.

## 1. Introduction

G protein-coupled receptors (GPCRs) are the largest class of cell-surface receptors (membrane proteins) [1] and are encoded by more than 800 genes in the human genome [2]. Over 50% of the targets of current United States Food and Drug Administration (FDA)-approved drugs are integral membrane proteins [3]. The majority of these drug targets fall within four well-studied protein superfamilies (GPCRs: 30%; voltage-gated ion channels: 8%; ligand-gated ion channels: 7%; and transporters: 7%) [3]. GPCRs form the largest portion as they are involved in a wide range of physiological processes including vision, taste, smell, inflammation, cell recognition, pheromone signaling, and many more. Various molecules such as hormones, lipids, peptides, and neurotransmitters exert their biological effects by binding to GPCRs coupled to heterotrimeric G-proteins, which are highly specialized transducers able to modulate diverse signaling pathways [4].

Different classification systems have been developed to sort out the superfamily of GPCR proteins. The most frequently used system is the International Union of Basic and Clinical Pharmacology (IUPHAR) [5], which divide GPCRs into six major classes, A (rhodopsin-like), B (secretin receptor family), C (metabotropic glutamate), D (fungal mating pheromone receptors), E (cyclic AMP receptors), and F (frizzled/smoothened) [5]. The classes/families are further broken down into sub-family, sub-sub-family, and sub-types based on their sequence homology and functional similarity [5,6]. It is common that a single ligand can bind to multiple GPCRs in the same family, but it is much less likely to see the same ligand binding to multiple GPCRs across different families (see Appendix A). Yet, one can envision the benefit of understanding the binding of the same ligand to GPCRs from different families in, for example, helping the process of drug repurposing and anticipating side effects. This motivated our attempt to look for common characteristics among GPCRs from different families that enable them to bind to the same ligand.

GPCR ligand binding is essential to many life processes in living organisms [7]. They are required for many proteins to function properly [8]. Interactions between GPCRs and their ligands are prerequisite for signal transduction, immunoreaction, and gene regulation [9]. GPCR ligand interaction studies are not only important for understanding the mechanisms of biological regulation, but also provide a theoretical basis for the design and discovery of new drug targets [9]. While GPCRs bind to G proteins inside the cell, they bind to a variety of extracellular ligands that include ions, biogenic amines, peptides, hormones, growth factors, lipids, and photons [10]. For example, galanin, an endogenous ligand for the GPCR galanin receptor type 2 (GALR2), plays an important role in epilepsy [11,12]. In humans, the initial phase of visual perception includes photon retention by four distinctive visual pigments [13]. These visual pigments include the apoprotein opsin covalently bound to the chromophore 11-cis-retinal (11CR), a vitamin A derivative that functions as an inverse agonist locking the photoreceptor opsin protein in its inactive state [13]. Similarly, inhaled selective β2-agonists (e.g., salbutamol, formoterol, indacaterol, etc.) are usually used in the treatment of obstructive airway diseases such as asthma [14]. These drugs bind to the β2-adrenoceptor (β2-AR) and causes the activation of certain G-proteins and subsequent generation of cyclic adenosine monophosphate (cAMP) in airway smooth muscles leading to bronchodilation [14]. GPCRs are also known to have crucial effects on tumor growth and metastasis [4,15].

Understanding how GPCRs bind to their ligands can help to identify characteristics that influence GPCR ligand interactions, which, in turn, help to fine-tune computational tools in terms of selecting feature sets for predicting GPCR ligand binding. It can also provide insights towards deciding on which computational tools to use. For example, as in [16], a novel protein-ligand binding prediction method, which makes use of the local and global structure of a ligand and amino acid motif sequence of a GPCR, has been proposed. Their method infers hidden properties of good ligand-receptor binding, which are encoded as a random forest classifier. Ciancetta et al. [17] have demonstrated that molecular dynamics can be used to account for critical aspects such as a realistic microenvironment for GPCRs, GPCR flexibility, and water molecule-mediated interactions that might play significant roles in ligand binding, which are neglected by molecular docking. GPCRs’ inherent flexibilities allow them to function through molecular interactions by changing their structural conformations in response to the presence of other molecules or variations in the environment [18].

It has been revealed through the analysis of protein-ligand complexes deposited in the Protein Data Bank (PDB) that most small organic molecules interact with specific pocket-like indentations on the surface of their target proteins. These surface regions are called binding sites or binding pockets [19]. It is now broadly realized that distant proteins might have comparable binding sites with capacities to recognize chemically similar ligands [19]. Various computational tools have been developed to evaluate the similarity of binding sites in proteins. These include PocketMatch [20], eF-seek [21], Patch-Surfer [22], eF-site [23], and CavBase [24] that use alignment-free techniques [19] and the alignment-based tools such as SiteEngine [25], Alignment of Pockets (APoc) [26], Sequence Order-Independent Profile-Profile Alignment (SOIPPA) [27], and Graph-based Local Structure Alignment (GLoSA) [28].

Structural similarities in binding poses among small molecules have proved important in docking. RosettaLigandEnsemble (RLE) was developed based on structural similarities in binding poses among small molecules that bind to one binding pocket [29]. RLE was found to generate more consistent docking results within a congeneric series and rescue the unsuccessful docking of individual ligands [29]. In [30], the authors have demonstrated that in 14% of related ligand pairs solved in complex with the same protein partner, their binding mode changes upon chemical elaboration of the smaller ligand of the pair. They have shown that simple structure-based modeling is more effective for identifying chemical substitutions that alter the binding mode for these pairs of ligands [30]. Some ligand pairs change binding mode because the added substituent would irreconcilably conflict with the receptor in the original pose, whereas others change because the added substituent enables new, stronger interactions that are available only in a different pose [30]. In the quest to understand ligand binding mechanisms in the human opsins, which are GPCRs responsible for light absorption in rod and cone cells in the retina, a secondary binding site for the ligand retinal has been identified [13]. Comparing the primary and secondary binding sites of rhodopsin in particular, we observed that the pose and orientation of the retinal docked at the secondary retinal-binding site (cyan) were different from that crystallized in the primary binding site from PDB (purple) (Figure 1). When compared using APoc, the two binding sites have a Pocket Similarity (PS)-score of 0.284 and an RMSD of the pocket alignment to be 2.40 Å.

Given all the information described above, our aim is to gain more insights about the sequence and structural features of GPCRs from different families that enable them to bind to the same ligand. In particular, we wanted to help answer these questions:(1)Do the GPCRs that bind to the same ligand share any conserved sequence motifs? Are they locally similar in terms of their 3D structures?(2)For GPCRs that bind to the same ligand, how similar are their binding pockets in terms of sequence and structure? Which residues of the GPCR interact with which atoms of the ligand?(3)For the ligands binding to human GPCRs from different families, do they bind with the similar poses and affinities?

## 2. Materials and Methods

This section presents methods that were used to assess and answer the questions raised above. Figure 2 shows the workflow of this study.

### 2.1. Dataset Collection

We collected publicly available datasets on 3D structures and sequences of GPCRs for structural comparison and molecular docking, and sequence motif search, respectively. In addition, the GPCR sequences, we gathered data on the start and end of their regional sequences (the N-terminal, extracellular loops, intracellular loops, the seven transmembrane helices, and the C-terminal).

Starting with confirmed GPCRs in the GPCR-PEnDB database [6], information on GPCR ligand interactions was gathered, compiled, and restructured to facilitate computational analyses, and to determine ligands that bind to GPCRs across three different IUPHAR [33] families. The interaction data were gathered from IUPHAR, BindingDB [34], and GLASS [35]. The dataset contains a total of 1,061,462 GPCR-ligand interactions with information on ligands and the binding affinities (Ki, Kd, IC50, EC50, pKB, pKi, pKd, pIC50, pEC50).

In addition, it contains information on each ligand’s SMILES (Simplified Molecular Input Line Entry Specification—a linear notation for describing chemical structures), affinity relations, and InChIKey (International Chemical Identifier compact hashed code), potency, activity, inhibition, and action (i.e., agonist, full agonist, partial agonist, antagonist, inverse agonist, biased agonist, etc.).

Twelve ligands that bind to a mix of GPCRs within the same family, of different families, and also to non-GPCRs with protein-ligand complexes on PDB (Appendix A) were gathered as positive controls for the various docking analysis. The 12 ligands were selected to have GPCR representations from the different IUPHAR families. However, not all the families had representations for the lack of GPCR ligand complexes on PDB (Appendix A).

### 2.2. Motif Search

In the course of the exploratory data analysis, we observed that there were some ligands that bind to GPCRs of two or more distinct IUPHAR families. As a result, we decided to perform a motif search across the GPCRs these ligands bind to, to ascertain whether there are significant motifs that go across different IUPHAR families. We focused on ligands that bind to GPCRs of three distinct IUPHAR families. In doing this we used Multiple Expression motifs for the Motif Elicitation (MEME) system to search for motifs. MEME works by searching for repeated, un-gapped sequence patterns that occur in the protein sequences provided by the user [36,37,38]. We took three approaches in our motif search: (1) search on the entire sequence of the GPCRs; (2) search on regional sequences (the N-terminal, extracellular loops, intracellular loops, the seven helices, and the C-terminal) of the GPCRs; and (3) search on modified regional sequence (that is, start the regional sequence five amino acids before the actual start of the regional sequence and/or end the regional sequence five amino acids after the actual end of the regional sequence) of the GPCRs.

### 2.3. Structural Comparison

We performed pairwise 3D structural comparisons of the GPCRs that bind to the same ligand and assessed the comparisons based on the RMSD (root mean square deviation) value of the alignment. The comparisons were done using Flexible structure AlignmenT by Chaining Aligned fragment pairs allowing Twists (FATCAT) [39,40], which allowed us to perform both rigid and flexible 3D alignment. We made use of jFATCAT (Java port of FATCAT) provided by PDB. We performed a rigid FATCAT and a flexible FATCAT. The rigid FATCAT uses a rigid-body superposition to align the two structures whereas the flexible FATCAT introduces ‘twists’ between different parts of the proteins that are superimposed independently [39,40,41]. For GPCRs that had multiple structures deposited in PDB, we selected one of their PDB IDs with the longest protein sequence of the GPCR. In other words, the one with protein sequence length closer to the actual GPCR sequence length was chosen for 3D structural comparison. As part of the results, the output of FATCAT is the 3D structures of portions of the GPCRs that were found to be 3D structurally similar. These portions were used in further analysis (see Section 2.9).

### 2.4. Binding Pocket Prediction

Three-dimensional structures of the GPCRs under consideration were downloaded from PDB and cleaned of any unwanted molecules including ligands not under study in this work and molecules used to aid the determination of the 3D structure of the GPCRs. They were then submitted to the metaPocket [42] website for binding pocket prediction. The metaPocket webserver provides a consensus result by combining results from LIGSITE^CSC^ [43], PASS [44], Q-SiteFinder [45], SURFNET [46]. All predicted binding pockets were considered for pocket comparison and for molecular docking.

### 2.5. Binding Pocket Comparison

All binding pockets predicted by metaPocket for each of the GPCRs were compared against those of the GPCRs they share the same ligand with. The comparisons were performed using APoc [18] which implements iterative dynamic programming and integer programming to calculate the optimal alignment between a pair of binding sites considering the secondary structure and fragment fitting. APoc provides a scoring function called the Pocket Similarity (PS)-score, which quantifies the pocket similarity between two given pockets based on their backbone geometry, orientation of side chains, and chemical matching of aligned pocket residues [18,24]. APoc can be applied to both experimentally determined and computationally predicted ligand binding sites [25]. The scoring function of APoc takes into consideration the chemical similarity of the aligned amino acids of the pockets in comparison [25], as a result a separate chemical similarity score of the predicted binding residues of the pockets in comparison was not necessary. APoc was chosen over others for its good performance and ease of comparing multiple pockets in one run.

### 2.6. GPCR Ligand Docking

The ligands were docked using AutoDock vina [47] into all the predicted binding pockets for the GPCRs they are known to bind to. This was done because at the time of this study there were no known 3D complexes of the ligands bound to the GPCRs on Protein Data Bank (PDB) [48], therefore we had no information of where and how the ligand binds to the GPCRs. Six ligand modes (poses and conformations) were generated, and only the ligand mode (pose and conformation) with the most negative value for the binding affinities (i.e., strongest binding) were retained for further analysis. We assessed the correlation between the PS-scores provided by APoc and the absolute difference of the binding affinities. Note that the smaller this absolute difference, the more alike the two pockets are in terms of their binding strengths for the same ligand. A negative correlation would imply that the PS-score is a good predictor of similarity in binding affinities.

### 2.7. Ligand Binding Pose and Conformation

After docking the ligands to their respective GPCRs, we aligned the docked ligand 3D structures using the PyMOL structural alignment method *align*, which performs a sequence alignment followed by a structural superposition [49]. This allowed us to assess the conformation of the docked ligands using the root mean squared deviation (RMSD) from the alignment. The Pearson correlation between the pocket similarity score (PS-Score) and the RMSD was assessed. The ligand poses were visually examined using PyMOL.

To assess the reliability of the docking results of the study data, we aligned the ligands of the pairs of pockets as deposited on PDB and recorded the RMSD from the alignment, RMSD_Actual_, of the control data. We then docked the same ligands into the same pockets as found on PDB and then aligned the ligands docked into the pairs of pockets of the control data. The RMSD from the alignment of the docked ligands were recorded, RMSD_Docked_. We assessed the reliability of the docking results by measuring the difference between the two sets of RMSD data of the control data and the relationship between PS-scores and RMSD_Actual_.

### 2.8. Protein Ligand Interaction

After the ligands have been docked into all the predicted binding pockets, we use LigPlot+ [50] to determine which residues of the GPCRs interact with which atoms of the ligand. LigPlot+ generates schematic 2D representations of protein–ligand complexes with colored outputs, postscript files containing information on intermolecular interactions and their strength, including hydrogen bonds, hydrophobic interactions, and atom accessibilities [50]. LigPlot+ was used to ascertain the interaction of retinal with the lysine amino acid of rhodopsin as proposed by Srinivasan et al. [30]. This interaction was detected within a threshold of 5 Å. Based on this, we chose hydrogen bonds detected within 5 Å by LigPlot+. We gathered data on the number of same residues across the pockets we are comparing that interact with the ligand from the LigPlot+ results output.

### 2.9. Predicted Pocket and 3D Structural Similarity Comparison Overlap

After the proteins that bind the same ligand have been compared based on their 3D structure, we compared the portions of the proteins that were similar from the pairwise 3D structure comparison (see Section 2.3) to their own predicted pockets. For example, let the 3D structures of the GPCR with PDB ID 3G04 be *A*, and that of the GPCR with PDB ID 7LCK be *B*. Let *P_A_* be the part of *A* that was found to be 3D structurally similar to *B*, likewise, *P_B_* be the part of *B* that was found to be 3D structurally similar to *A*. Let *PK*_(*A*,*i*)_, *i* = 1, 2, …, *m* be the predicted pockets of *A* and *PK*_(*B*,*j*)_, *j* = 1, 2, …, *n* be the predicted pockets of B. Each *PK*_(*A*,*i*)_, *i* = 1, 2, …, *m* was compared with *P_A_* and each *PK*_(*B*,*j*)_, *j* = 1, 2, …, *n* was compared with *P_B_*. A score was calculated for each comparison using the formula below:(1)$$S_{\left(A,i\right),B}=\frac{N\left(aaP_{A}\cap^{\;}aaPK_{\left(A,i\right)}\right)}{N\left(aaPK_{\left(A,i\right)}\right)},\;S_{\left(B,j\right),A}=\frac{N\left(aaP_{B}\cap^{\;}aaPK_{\left(B,j\right)}\right)}{N\left(aaPK_{\left(B,j\right)}\right)},$$
where *S*_(*A*,*i*),*B*_ is the overlap score for the ith pocket of *A* with *P_A_* when *A* is compared with *B*, *S*_(*B*,*j*),*A*_ is the overlap score for the jth pocket of *B* with *P_B_* when *B* is compared with *A*, *aaP_A_* is the set of positions of the ordered amino acids of *P_A_*, *aaP_B_* is the set of positions of the ordered amino acids of *P_B_*, *aaPK*_(*A*,*i*)_ is the set of positions of the ordered amino acids of *PK*_(*A*,*i*)_, *aaPK*_(*B*,*j*)_ is the set of positions of the ordered amino acids of *PK*_(*B*,*j*)_, *N*(•) is the count of the positions of the amino acids. The code for the scoring is included as part of the Appendix A. Overlap scores were calculated for both cases of using flexible 3D structural comparisons and rigid 3D structural comparisons (see Section 2.3). An average score for the flexible and the rigid cases were obtained for each pocket. The average scores were summed together for each pair of pockets compared using the formula below:(2)$$S_{\left(A,i\right),\;\left(B,j\right)}={\bar{S}}_{\left(A,i\right),B}+{\bar{S}}_{\left(B,j\right),A},$$
where ${\bar{S}}_{\left(A,i\right),B}$ is the average overlap score for the flexible and the rigid case of pocket *PK*_(*A*,*i*)_, ${\bar{S}}_{\left(B,j\right),A}$ is the average overlap score for the flexible and the rigid case of pocket *PK*_(*B*,*j*)_, and *S*_(*A*,*i*), (*B*,*j*)_ is the sum of average overlap scores for the pair of pockets, *PK*_(*A*,*i*)_ and *PK*_(*B*,*j*)_, compared. The summed score (Equation (2)) was then analyzed to determine their Pearson correlation with the PS-score. This was done to check if the parts of the proteins that are 3D structurally similar are also locations for pockets.

### 2.10. Pockets Electrostatic Properties

We calculated Molecular Surface Weighted Holistic Invariant Molecular (MS-WHIM) scores (made up of three values, *x*, *y*, *z*) [51,52] of the amino acids of each of the pockets using the R package *Peptides* [53]. MS-WHIM scores are obtained from electrostatic potential properties derived from the 3D structure of the 20 natural amino acids as described in reference [51,52]. We calculated the Chebyshev distance between the MS-WHIM scores of any two pockets under comparison. For example, let *PK*_(*A*,*i*)_, *i* = 1, 2, …, *m* be the predicted pockets of GPCR A and *PK*_(*B*,*j*)_, *j* = 1, 2, …, *n* be the predicted pockets of GPCR B that are under comparison. For each pocket *PK*_(*A*,*i*)_ and *PK*_(*B*,*j*)_, we calculate MS-WHIM scores of the amino acids that form the pockets. These scores ${\left(x,y,z\right)}_{PK_{\left(A,i\right)}}$ and ${\left(x,y,z\right)}_{PK_{\left(B,j\right)}}$ for the pockets we are comparing were seen as points in the 3D-plane, and the distance D between the two points was calculated using Chebyshev distance.
(3)$$D\left[{\left(x,y,z\right)}_{PK_{\left(A,i\right)}},\;{\left(x,y,z\right)}_{PK_{\left(B,j\right)}}\right]=\max\left(\left|x_{i}-x_{j}\right|,\left|y_{i}-y_{j}\right|,\left|z_{i}-z_{j}\right|\;\right).$$

The Chebyshev distance was calculated for each pair of pockets we were comparing. The distances were later used to determine the Pearson correlation between the PS-score and MS-WHIM. To visualize the electrostatic properties, electrostatic surface properties of the pockets were generated using APBS and PDB2PQR [54] plugins for PyMOL [49]. The data for this study is included in the Appendix A.

## 3. Results and Discussion

This section presents results obtained from the methods described above.

### 3.1. 3D Structures of GPCR

There were 817 atomic-level 3D GPCR structures related to 161 distinct GPCRs, of which 107, 27, 16, 2, 0, 9, and 0 are in Class A, B, C, D, E, F, and T2R, respectively, on PDB at the time of the study (Figure 3A). Generally, the number of new 3D structures of GPCRs is increasing rapidly by the year (Figure 3B).

### 3.2. GPCR Ligand Binding

The GPCR-ligand binding data have revealed that there are ligands that bind to human GPCRs of multiple IUPHAR families. There were 11 ligands that bind to members of three different IUPHAR families (Appendix A). These 11 ligands had 106 interactions with human GPCRs, involving 69 unique GPCRs of which 42 had entries in RCSB PDB. IUPHAR classifies three of these ligands as synthetic organic based on their nature (Appendix A).

It should be noted that out of the 11 ligands, 3 were found to bind to a rather large number of GPCRs (e.g., ligand FQUA bound to 25 Class A, 6 Class B, and 2 Class C GPCRs), each of which may have several binding pockets. Those three GPCRs were estimated to generate over 30,000 binding pocket pairs in total. To reduce the number of pairwise pocket comparisons, we excluded the three ligands from this study and focused only on the remaining eight ligands for which a total of about 990 pocket pairs were compared to produce the results in Section 3.5 below.

### 3.3. Conserved Motifs

We performed this analysis on the full sequence, regions of the sequence (extracellular loops, intracellular loops, and the seven transmembrane helices), and modified regions of the sequence (i.e., adding five amino acids either at the beginning or at the end of the regional sequence or at both ends; this was done only to the extracellular loops). The regions of the GPCR sequences were labeled *E_i_*, *i* = 1, 2, 3: extracellular loops; *I_i_*, *i* = 1, 2, 3: intracellular loops; N-terminal; C-terminal; *H_i_*, *i* = 1, 2, 3, 4, 5, 6, 7: transmembrane helices, and Full: full GPCR sequence. Table 1 lists the conserved motifs with E-value < 0.1. Since GPCRs are classified into families based on their sequence and function [55], we expected that only a few of the GPCRs under study would share conserved motifs, and that was the case in our analysis (Table 1). The ligand names are an abbreviation of the International Chemical Identifier compact hashed code (InChIKey) (Appendix A).

### 3.4. Structural Comparison

Structural comparisons were done on the entire GPCR structures if the GPCRs involved are from the same IUPHAR family, whereas regional comparisons were done when the GPCRs involved are from different IUPHAR families. In the column “GPCR UniProt ID and IUPHAR Class” of Table 2, for example P16473.A, the characters before the ‘.’ represents the UniProt ID of the GPCR, whereas the character after the ‘.’ represents the IUPHAR families. Similarly, in the column ‘PDB ID and Chain ID’ of Table 2, for example 3G04.C, the string before the ‘.’ represents the PDB ID of the GPCR, whereas the character after the ‘.’ represents the chain ID. The chain IDs were chosen based on the length of the sequence, i.e., the chain with the longest sequence was chosen for the comparison.

We observed that over 75% of the pairs compared in Table 2 have both flexible and rigid sequence similarities below 30%, implying they are unlikely to have similar structures. Even for the pair P35462.A and P14416.A with flexible and rigid sequence similarities as high as 88% and 64%, respectively, their rigid RMSD is 9.63, again indicating that there are differences in their structures. The results in Table 2 suggest that two GPCRs, even when they are quite dissimilar in their sequences and structures, can bind to the same ligand. Some of the GPCRs that these ligands bind to have no 3D structures on PDB, and as a result those GPCRs were excluded from further analysis as in the case of AJLF, IKSH, and NKOP (Table 2).

### 3.5. Binding Pocket Comparison and GPCR Ligand Docking Relationship

Since there was no known complex of the ligands bound to the GPCRs on PDB, as a result we had to perform pocket comparison of all possible pairs of pockets; we decided to analyze the relationship between the pocket comparison similarity score, PS-score, and the absolute difference of the binding affinities when the ligand is docked into the pocket. Despite not having a ligand bound to the GPCR complex on PDB, we observed that, on the average, PS-score over all pairs of pockets of GPCRs that bind the same ligand was 0.287 and above. This value is comparable to the PS-score of 0.284 obtained for the two retinal-binding sites in rhodopsin as mentioned earlier in the Introduction section. Moreover, among the maximum PS-score values in Table 3, the lowest was 0.349, suggesting that the GPCRs binding to the same ligand would contain similar binding pockets with PS-score no less than 0.349.

We also observed a significant negative correlation between the PS-score and the absolute difference of the binding affinities of the pockets in comparison (Table 4). These results suggest that binding pockets that are similar share similar binding affinity, thus increasing PS-score is associated with similar binding affinity. By extension, the binding pocket residues could be chemically similar.

To verify these results, we used the small control dataset with known interactions and performed binding pocket predictions. We observed that 11 out of the 12 ligands had their actual binding site of the ligands as identified on PDB as part of the predicted pockets. The control dataset also showed a significant negative correlation (r = −0.4547, *p*-value = 0.01718) between the PS-score and the absolute difference of the binding affinities of the pairs of pockets across the ligands with a minimum PS-score of 0.114 (Table 3).

### 3.6. Ligand Binding Pose and Conformation

Figure 4 shows a plot of the RMSD_Actual_ and RMSD_Docked_ of the aligned ligands of the pairs of pockets as found in complex with the proteins deposited on PDB and when docked into the same binding site (see Section 2.7). We observe from Figure 4 that the two plots exhibit a similar pattern for the control dataset. Additionally, a paired sampled *t*-test on the RMSD_Actual_ and RMSD_Docked_ revealed no significant difference in the mean RMSD_Actual_ (1.2414 ± 0.7654) and RMSD_Docked_ (1.3900 ± 0.5060) (t = −1.5849, df = 26, *p*-value = 0.1251) for the control dataset. These results are good indications for the reliability of the docking results.

To study the ligands’ poses and conformations when binding to their target GPCRs, we chose the pocket pairs with the highest PS-scores, which are all greater than 0.28 (reference PS-score of the two rhodopsin-retinal binding pockets). Generally, we observed varying poses and conformation across the pairs of pockets. A few representative examples were selected to demonstrate the differences in pose and in conformation after docking. The ligand AJLF binds with different poses to pocket 1 of 3G04 and pocket 5 of 7LCK (Figure 5A,B). However, they share very similar conformation (Figure 5C). Similar observations were made for the ligand CLQV (Figure 5D–G). Nevertheless, in the case of the ligand DTZD, there were noticeable differences in the bound conformations of the ligand for the three pockets (Figure 5K). Like in the case of AJLF and CLQV, DTZD binds to pocket 2 of 3MQ4, pocket 4 of 7LCK, and pocket 2 of 7LJC with different poses (Figure 5H–J). However, the Pearson correlation analysis reveals a significant negative correlation (r = −0.0737, *p*-value = 0.02037) between the PS-score and the RMSD of the aligned docked ligands (see Section 2.7) across all pairs of pockets. Similar results were obtained for the control dataset, showing a significant negative correlation (r = −0.5437, *p*-value = 0.0034) between the PS-score and the RMSD_Actual_ across all pairs of pockets. These results indicate that the more similar the pockets are, the more likely the ligand conformation will be the same.

### 3.7. Protein Ligand Interaction and Pocket Electrostatic Properties

We observed that the ligands tend to bind to similar pockets that share similar residues. The Pearson analysis reveals a significant positive correlation (r = 0.0649, *p*-value = 0.04135) between the PS-score and the number of same residues across all pairs of pockets that interact with the ligands. This suggests that the more similar the pockets are, the more likely a ligand binding to the pockets will interact with the same residues across the pockets. For example, the ligand AJLF interacts with serine in pockets 1 and 5 of the GPCRs 3G04 and 7LCK, respectively, (Figure 6A).

Generally, we observed that pockets that are similar are also similar in electrostatic properties (Figure 6B), thus increasing PS-score is associated with similar electrostatic properties. The Pearson correlation analysis within the studied ligands revealed mostly negative correlations between the PS-score and the distances obtained from the MS-WHIM scores of the compared pockets (see Section 2.10). Although only half of the correlations were statistically significant (Table 5), this is an indication that similar pockets may be similar in electrostatic properties and should be further investigated with more ligand pairs in the future.

### 3.8. Predicted Pocket and 3D Structural Similarity Comparison Overlap

We found that some of the predicted pockets of the GPCRs overlap with portions of the GPCRs that were found to be similar from the pairwise 3D structure comparison (see Section 2.3 and Section 2.9). There is a significant positive correlation (r = 0.2921, *p*-value = 2.2 × 10^−16^) between the PS-score and the overlap scores across all pairs of pockets. This suggests that the more similar the pockets are, the more likely the pockets are found in a region where the two proteins are structurally similar in their 3D structures.

## 4. Conclusions

We set out to computationally uncover and understand the binding of a single ligand to GPCRs from three different families. As expected, relatively few of such GPCRs of different families share conserved sequence motifs or global structural similarities. However, many share local 3D structural similarities and similar binding pockets. Moreover, the more similar the pockets are, the more likely their binding ligands will interact with the same residues across the pockets, with the same ligand conformation, and similar binding affinities across the pockets. In addition, the more similar the pockets are, the more likely the electrostatic properties of the pockets will be similar, and the more likely the pockets are found in a region where the two proteins are structurally similar in their 3D structures. These findings can be taken advantage of to further develop protein function inference, drug toxicity prediction, and discovery of unwanted cross reactivity to speed up the process of drug repurposing and new drug development.

## Figures and Tables

**Figure 1 biomolecules-12-00863-f001:**
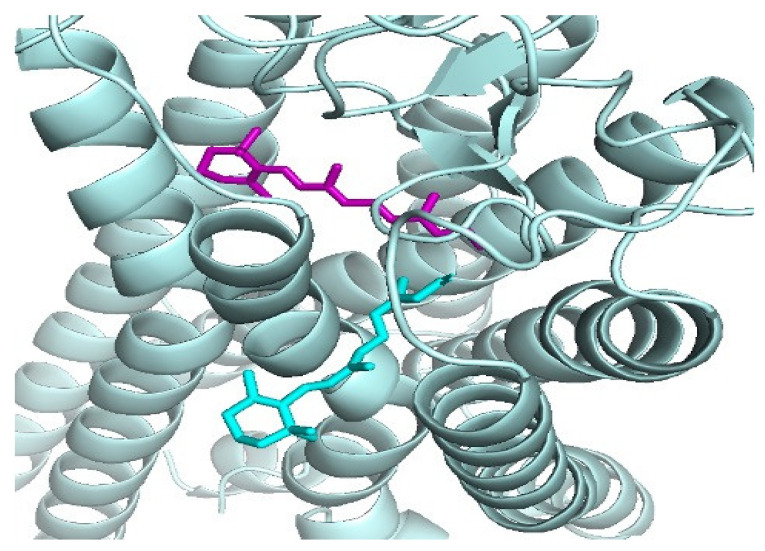
The docked retinal at the putative secondary retinal-binding site (cyan) proposed in [31] and the crystal complex of the retinal (purple) is bound with rhodopsin, PDB ID: 2X72, [32].

**Figure 2 biomolecules-12-00863-f002:**
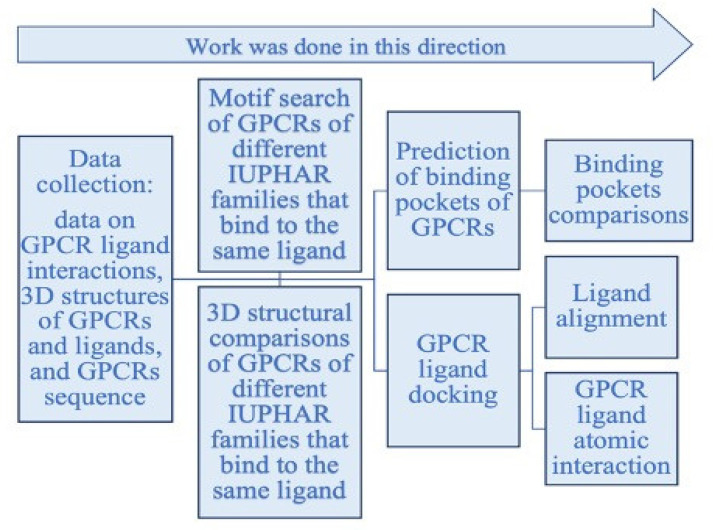
Workflow of the project.

**Figure 3 biomolecules-12-00863-f003:**
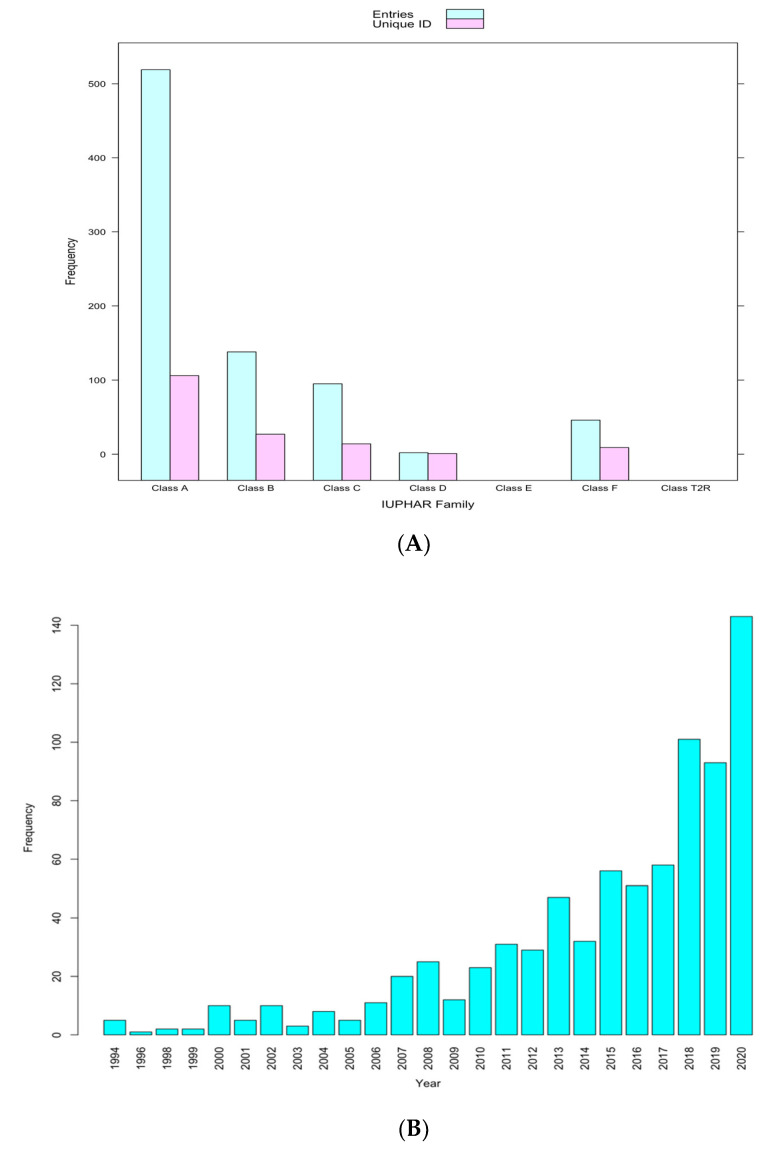
(**A**) Frequency distribution of GPCRs 3D structures in PDB classified by family. (**B**) Frequency distribution of new GPCR 3D structures added by year.

**Figure 4 biomolecules-12-00863-f004:**
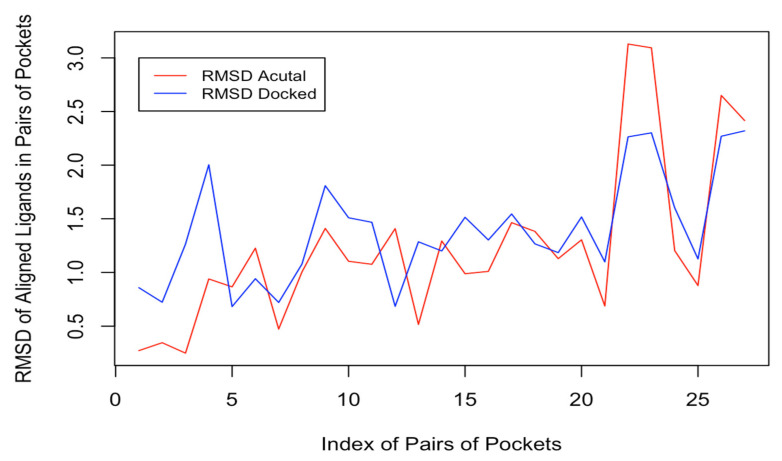
A plot of the RMSD_Actual_ and RMSD_Docked_ pairs of the pockets.

**Figure 5 biomolecules-12-00863-f005:**
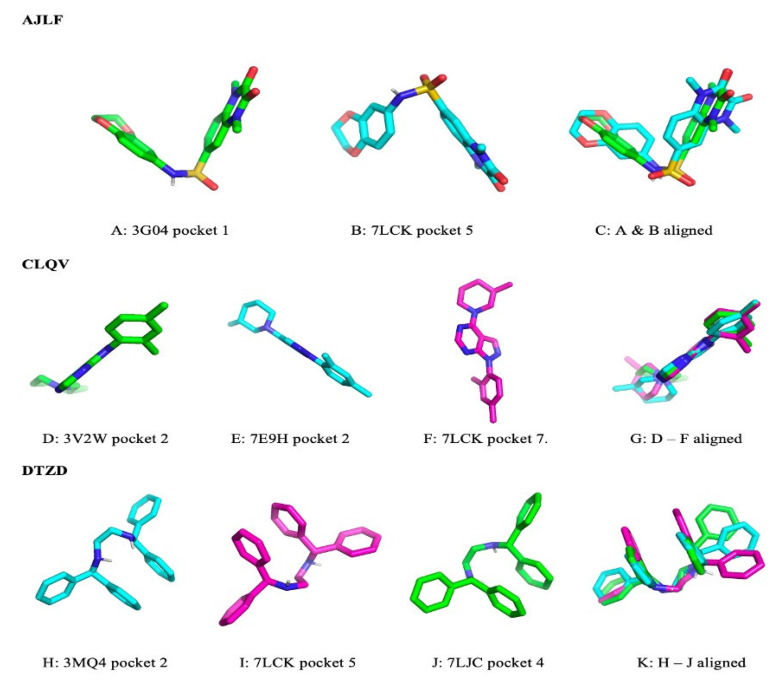
Ligand binding poses and conformations across the different GPCRs binding pockets.

**Figure 6 biomolecules-12-00863-f006:**
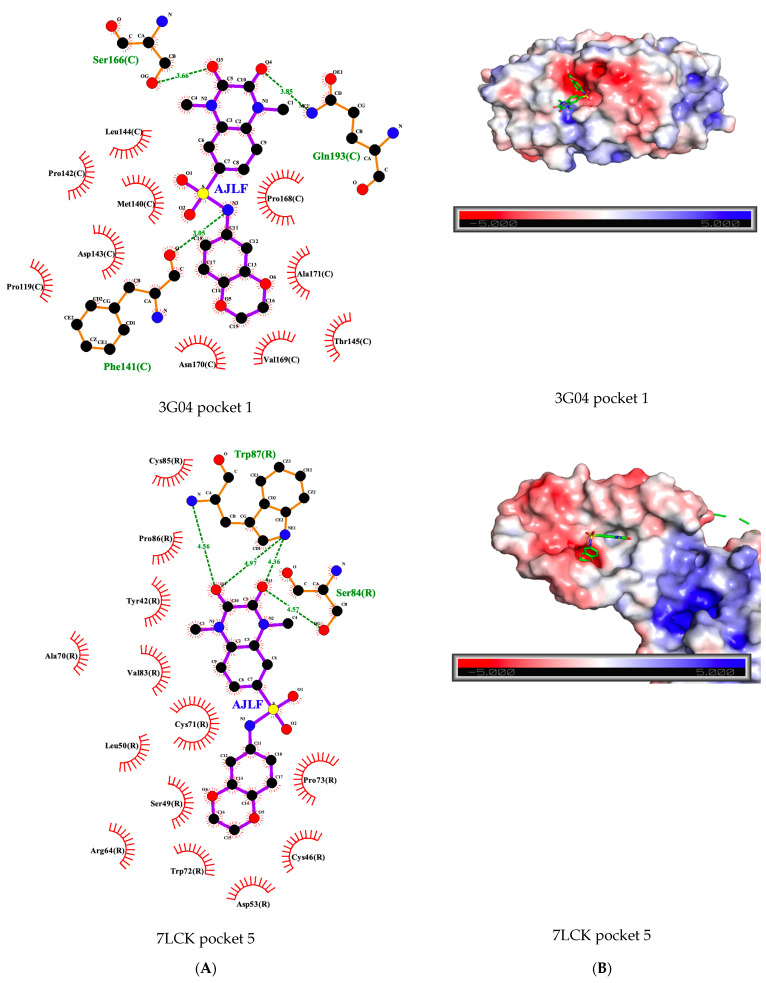
(**A**) AJLF GPCR interaction in the pocket. (**B**) Pocket electrostatic properties with AJLF docked into the pocket.

**Table 1 biomolecules-12-00863-t001:** Conserved motifs of human GPCR sequences across three IUPHAR families found by the MEME system.

Ligand	Region	UniProt ID	Family	Motif & E-Value
NKOP	Full	P43220	B	**QHQWD** 4.20 × 10^−2^
		Q13255	C	
XLWJ	N−terminal	Q03431, P43220	B	**GHVYRKCDANGSW** 5.50 × 10^−2^
		Q9UBS5	C	
	*I* _2_	Q6W5P4, P21728, P16473	A	**DRYHAITYPM** 7.60 × 10^−2^
		O75899	C	
	*H* _7_	Q6W5P4, P21728, P16473	A	**NSALNPIIYC** 5.10 × 10^−2^
		P43220, Q03431	B	
YKMS	C−terminal	P14416, P35462, P21917	A	**EFRKAFLKILRC** 2.10 × 10^−4^
		P43220	B	

**Table 2 biomolecules-12-00863-t002:** Pairwise structural comparison of human GPCRS binding to the selected ligands using jFATCAT.

Ligand	GPCR UniProt ID and IUPHAR Class	PDB ID and Chain ID	RMSD (Å)		Sequence Similarity (%)	
			Flex	Rigid	Flex	Rigid
**AJLF**	P16473.A, P43220.B	3G04.C, 7LCK.R	2.92	4.87	19	17
**DTZD**	P43220.B, Q14831.C	7LCK.R, 3MQ4.A	3.63	10.53	20	16
	P43220.B, P21728.A	7LCK.R, 7LJC.R	3.19	3.19	33	33
	Q14831.C, P21728.A	3MQ4.A, 7LJC.R	3.88	10.72	19	15
**CLQV**	P21453.A, P43220.B	3V2W.A, 7LCK.R	2.84	4.94	27	21
	P21453.A, Q14833.C	3V2W.A, 7E9H.A	3.99	4.51	24	25
	Q14833.C, P43220.B	7E9H.A, 7LCK.R	4.37	6.44	24	24
**IKSH**	P16473.A, Q03431.B	3G04.C, 6FJ3.A	2.98	4.66	18	22
**NKOP**	P43220.B, Q13255.C	7LCK.R, 3KS9.A	4.47	8.08	18	17
**USZP**	P16473.A, P43220.B	3G04.C, 7LCK.R	2.92	4.87	19	17
	P16473.A, P41594.C	3G04.C, 6N52.A	3.06	6.23	15	18
	P43220.B, P41594.C	7LCK.R, 6N52.A	3.83	9.67	25	20
**XLWJ**	P16473.A, P43220.B	3G04.C, 7LCK.R	2.92	4.87	19	17
	P16473.A, Q03431.B	3G04.C, 6FJ3.A	2.98	4.66	18	22
	P16473.A, P21728.A	3G04.C, 7LJC.R	3.81	3.81	23	23
	P43220.B, Q03431.B	7LCK.R, 6FJ3.A	2.53	3.85	45	50
	P43220.B, P21728.A	7LCK.R, 7LJC.R	3.19	3.19	33	33
	Q03431.B, P21728.A	6FJ3.A, 7LJC.R	3.86	3.03	27	30
	O75899.C, P16473.A	6W2X.B, 3G04.C	3.16	6.04	16	22
	O75899.C, P43220.B	6W2X.B, 7LCK.R	2.92	5.69	24	22
	O75899.C, Q03431.B	6W2X.B, 6FJ3.A	3.20	4.60	24	22
	O75899.C, P21728.A	6W2X.B, 7LJC.R	3.03	3.15	25	26
**YKMS**	P35462.A, Q14416.C	3PBL.A, 5KZN.A	5.38	11.06	17	17
	P35462.A, P21917.A	3PBL.A, 5WIV.A	1.96	3.95	52	53
	**P35462.A, P14416.A**	**3PBL.A, 6CM4.A**	**2.15**	**9.63**	**88**	**64**
	P35462.A, P43220.B	3PBL.A, 7LCK.R	6.28	4.44	17	21
	Q14416.C, P21917.A	5KZN.A, 5WIV.A	5.42	6.76	19	19
	Q14416.C, P14416.A	5KZN.A, 6CM4.A	5.23	14.95	17	25
	Q14416.C, P43220.B	5KZN.A, 7LCK.R	5.64	9.41	15	23
	P21917.A, P14416.A	5WIV.A, 6CM4.A	2.72	8.72	52	50
	P21917.A, P43220.B	5WIV.A, 7LCK.R	3.01	4.88	21	23
	P14416.A, P43220.B	6CM4.A, 7LCK.R	3.49	4.44	32	31

**Table 3 biomolecules-12-00863-t003:** Summary statistics for PS-scores for all possible pairs of pockets for each ligand.

Ligand	Min	Mean	STD	Max
Control Data	**0.114**	0.487	0.206	0.901
AJLF	0.219	**0.287**	0.0374	0.372
CLQV	0.212	0.326	0.0554	0.528
DTZD	0.218	0.315	0.0498	0.466
IKSH	0.228	0.291	0.0309	**0.349**
NKOP	0.221	0.295	0.0375	0.369
USZP	0.210	0.295	0.0557	0.500
XLWJ	0.218	0.318	0.0570	0.508
YKMS	0.209	0.334	0.0802	0.733

**Table 4 biomolecules-12-00863-t004:** Correlation between PS-score and absolute difference of the binding affinities.

Ligand	Correlation	*p*-Value
XLWJ	−0.2776	1.29 × 10^−7^
AJLF	−0.7213	4.73 × 10^−5^
NKOP	−0.4732	6.23 × 10^−3^
IKSH	−0.3501	3.92 × 10^−2^
DTZD	−0.1947	4.55 × 10^−2^
USZP	−0.2056	7.68 × 10^−2^
CLQV	0.1162	0.238
YKMS	0.0318	0.609

**Table 5 biomolecules-12-00863-t005:** Correlation between PS-scores and Chebyshev distances of MS-WHIM scores for all compared pocket pairs by ligand.

Ligand	Correlation	*p*-Value
YKMS	−0.1936	0.0016
CLQV	−0.2872	0.0029
NKOP	−0.4035	0.0219
USZP	−0.2411	0.0372
AJLF	−0.1964	0.3466
XLWJ	−0.0314	0.5588
DTZD	−0.0306	0.7552
IKSH	0.0263	0.8805

## Data Availability

The data and major codes needed for this study are included in the Appendix A.

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
