# Peer review of "What Makes GPCRs from Different Families Bind to the Same Ligand?"

_biomolecules, 2022, doi:10.3390/biom12070863_

Round 1

Reviewer 1 Report

While accepting to review the paper I was immediately drawn to the premise of the paper. The authors ask a very interesting question, the answers to which I believe will be of tremendous interest to researchers working with GPCRs and ligand discovery in general.

The introduction, data selection, and methods employed are described and performed very well. However, I am not convinced that the authors have managed to address the questions they posed to a sufficient degree of clarity.

I will highlight some of the concerns I have regarding the research presented in the paper.

1) The authors chose 11 ligands that they found to bind to GPCRs belonging to three distinct classes based on binding data and selected 8 for further analyses. Why choose 8? Why not 5, 6, or 7? What is the rationale?

2) None of the ligands have structures determined in complex with GPCRs and the authors seem to completely rely on docking studies to ascertain their binding poses. Wouldn't it have helped to consider a few examples from ligands that bind to only one or two classes of GPCRs with structures available as a positive control? How reliable are the binding sites identified and the docking poses predicted? The approach of predicting based on earlier predictions without any solid experimental backing is concerning.

3) Why is Table 2 referenced in the paper after Table 1?

4) Ligand AJLF in Table 2 seems to bind only GPCRs from classes A and B. Then why is it considered to bind all three classes of GPCRs? Similar doubts arise for ligands IKSH and NKOP.

5) In figure 4, the authors show three ligands that bind to different predicted pockets in different GPCRs. How were these poses selected? Is it the pose with the best pose? How many different poses were generated for each site? What are their respective docking scores? How many conformations of the respective ligands were generated? Did all the poses adopt a similar conformation? How do the authors attest to the fact that the conformation space of the ligand and docked pose are sufficiently sampled? Why don't the authors report the docking scores or binding affinity values considered? I understand that the absolute difference of binding affinities is used? Why is that? Why no provide binding affinity values as well as the differences? I'd think that'd be more useful for the readers to make sense of the data presented.

6) The authors use Figure 4 to explain how the docked ligands adopt similar conformations despite binding to different sites in different GPCRs. How do they rule out poor sampling of ligand conformations? This is apparent in case of CLQV which is linear molecule. And why only show three molecules? Why not show all eight, even in the SI? And why are actual side-chains forming the binding pockets not shown while preferring LigPlot?

7) It would be useful for the readers if the authors provide all docked complexes generated in the study in the SI.

8) The authors ask very specific and pointed questions towards the end of the introduction section. Yet, the "Conclusions" section is vague and makes very few real conclusions. The statements made in the very brief "Conclusions" section are qualitative in nature and the results presented in the "Results" section do not, in my opinion, provide a solid quantitative basis to support.

I have several other concerns regarding the work but I will not put them forward here. I think the authors need to seriously introspect and iron out the flaws in the study.

Reviewer 2 Report

This manuscript uses currently published GPCR 3D structural data and computational predication of GPCR ligand binding to study how GPCRs from different families can bind to the same ligand. The manuscript is clearly written and should be interested to the readers in the GPCR field.

I have only two minor points. 

1.  On page 9, the first paragraph: It is not clearly stated why GPCRs have four intracellular loops and four extracellular loops? Most GPCRs have three intracellular loops and three extracellular loops. Does this include a loop formed in the C-terminus? This is very important as it will make it clear what I4 is. Another important structural feature of GPCRs which is not mentioned in the manuscript is that many GPCRs form helix 8 in their C-termini. In addition, I think regions of the sequence (line 303-304) should also include both termini.   

2. There are a number of language mistakes in the text and the authors need to proofread the manuscript.   

Author Response

Respones attached

Round 2

Reviewer 1 Report

I do not find the authors' responses satisfactory. There hasn't been much improvement in the matter presented in the paper. I agree that the authors have addressed some comments/suggestions. But it seems to me that the changes are tokenistic, and no real effort has been made to address the core issues I had pointed out in my earlier review.

My main criticism is that the title of the paper is promising and grabs the attention of the readers. It will draw in people but would end up disappointing them as the authors still make only vague conclusions. There is a serious disconnect between the promise from the title of the paper and the findings reported.

The authors have carried out a lot of analyses on a limited dataset. They do not wish to expand the dataset. The methods section provides a good roadmap as to how one may approach such a problem. The premise of the work is promising but the findings are not (at least in my view). If the authors are intent on publishing this work without further improvements, the least they can do is to make sure the title of the work is truly reflective.

Reviewer 2 Report

The authors have addressed my concerns.